# Interpretable multiple instance learning for hematologic diagnosis from peripheral blood smears

## Abstract

**Background** Accurate diagnosis of hematologic malignancies from peripheral blood smears (PBSs) requires integrating cellular morphology and composition across numerous white blood cells. Existing computational approaches predominantly automate single-cell classifications and do not provide holistic, slide-level diagnostic predictions.

**Methods** We present a framework that employs a high-performance cell-based encoder (DeepHeme) for feature extraction, integrated with our weakly supervised, attention-based multiple instance learning (MIL) model, termed CAREMIL (Cell AggRegation, Explainable, Multiple Instance Learning). Through comprehensive evaluations of leading image encoders and MIL architectures, the combination of DeepHeme and CAREMIL demonstrated superior performance on disease classification tasks. CAREMIL functions as a robust aggregation mechanism, consistently outperforming established slide-level MIL methods (gated MIL and Dual-stream MIL Network) across multiple encoder types. The most pronounced performance gains were observed with out-of-domain encoders, including ImageNet-pretrained and open-source pathology foundation models (UNI2 and Virchow2).

**Results** CAREMIL combined with DeepHeme achieves the highest diagnostic accuracy across acute myeloid leukemia (AML), myelodysplastic syndromes (MDS), and hairy cell leukemia (HCL), with AUROCs of 0.999, 0.891, and 0.945, respectively, and successfully identifies AML even in cases with minimal or absent circulating blasts. Attention values assigned by CAREMIL highlight diagnostically relevant cells and reveal disease-specific morphometric patterns, enabling biological interpretability and case-level insights. The framework remains resilient to individual cell misclassifications and does not require explicit cell-level supervision.

**Conclusions** These findings establish CAREMIL as an effective and interpretable MIL framework for hematologic slide diagnosis, extendable to bone marrow aspirates, cytology, and other liquid biopsy specimens, supporting a shift toward quantitative, morphology-informed hematologic diagnostics.

## Plain Language Summary

Computational models can be used to analyse images of cells and tissues taken from people with cancer. Most models are designed for images of solid tissue and do not base their analysis on the appearance of individual cells. This means they do not work so well on liquid samples such as blood samples. We developed a system to detect blood cancers from images of blood and found it worked better than models previously developed for solid tissue analysis. Our computational model could also be used by other researchers to discover additional diseases detectable from blood samples or be expanded to enable population-scale blood cancer screening.

Accurate diagnosis of hematologic malignancies from peripheral blood smears (PBSs) requires integrating both the morphology of individual white blood cells (WBCs) and the overall cellular composition[1–4]. Peripheral smears serve as a non-invasive, rapid "liquid biopsy", offering critical information about underlying bone marrow pathology. However, diagnosis often hinges on detecting subtle morphological abnormalities or shifts in cellular proportions, making interpretation highly dependent on expert review[5]. These challenges are particularly acute in cases of low circulating

disease burden or early-stage dysplasia, where abnormalities may escape detection through routine screening[6,7]. The clinical and computational workflow for PBS analysis is outlined in Fig. 1A. Recent computational efforts have focused primarily on automating aspects of PBS interpretation, including single-cell classification and cytometric quantification[8–10]. While these approaches improve efficiency, they fail to capture the integrative reasoning required for whole-slide diagnosis. In contrast to solid tumor pathology, where tiling approaches can leverage preserved tissue

✉e-mail: singis@mskcc.org

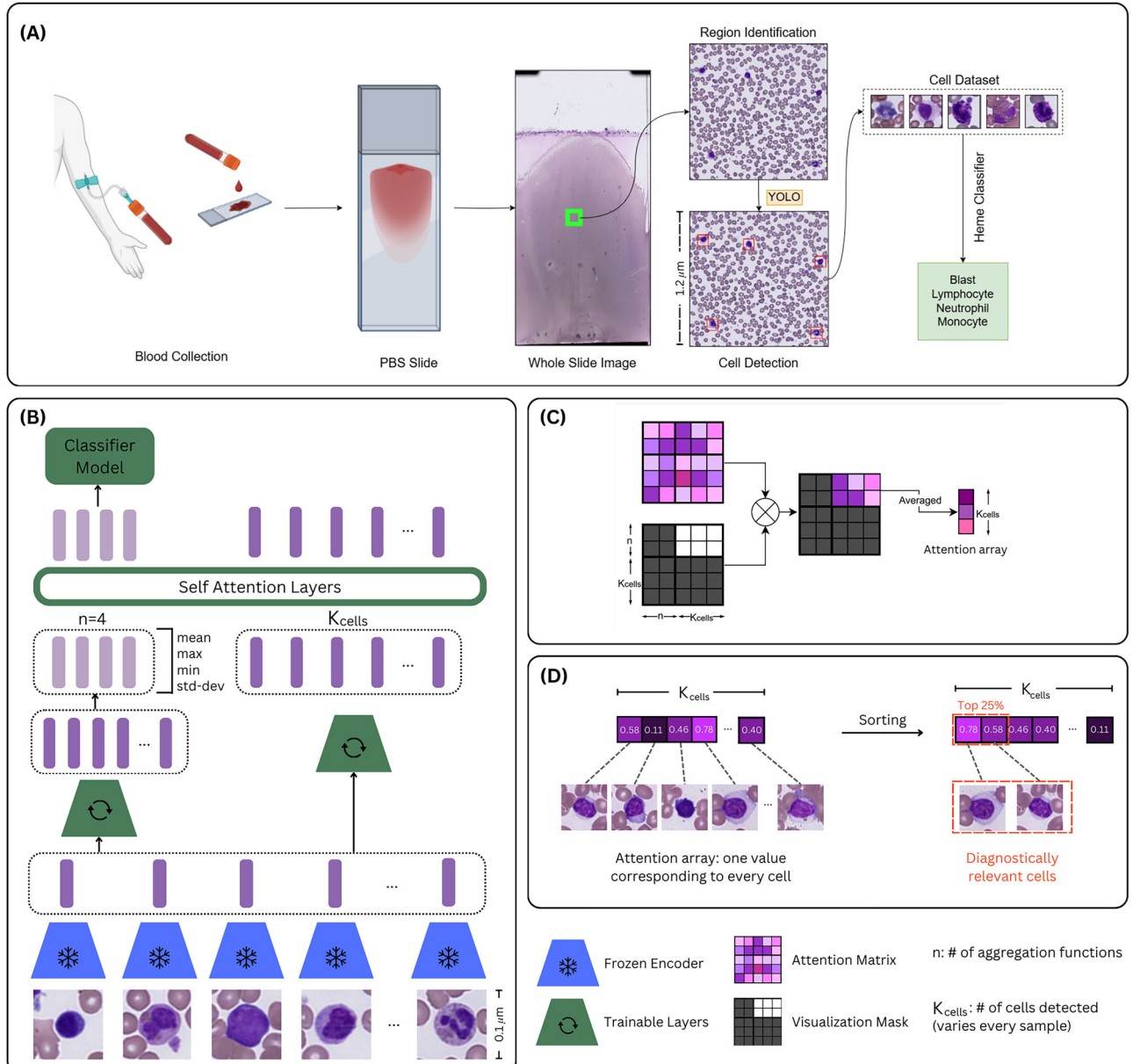

**Fig. 1 | Overview of CAREMIL.** CAREMIL is a deep learning pipeline designed for pathologists to diagnose and analyze liquid biosamples. **A** Data collection process. **B** Model architecture: The model is designed to take in a varying number of cells for every sample and make an explainable classification prediction. **C** Masked Attention Visualization. We isolate the part of the attention matrix that relates the aggregation functions to every cell in the input of the model. **D** Cell Ranking: cell ranking by prediction importance identifies the cells with highest and lowest diagnostic relevance to verify a decision. Created in BioRender. Vanderbilt, C. (2026) https://BioRender.com/e96v954.

architecture[11,12], hematologic specimens lack cohesive spatial organization[13], rendering traditional whole slide image (WSI) analysis methods poorly suited for blood smears, where proportions of cell types and single-cell morphology are central to diagnostic interpretation, rather than tissue architecture[14,15]. As a result, new frameworks are needed to model PBSs at the single-cell level while enabling slide-level diagnostic predictions. Multiple instance learning (MIL) offers a promising solution for weakly supervised learning in pathology[12,16–20], treating individual cells as instances and the PBS as a labeled bag. Attention-based MIL architectures have shown success in solid tumor WSIs by improving both performance and interpretability, but their adaptation to hematologic specimens remains limited. Prior work has applied MIL to blood smear images using general-purpose encoders without attention-based aggregation, but comprehensive strategies for integrating morphology and composition with domain-specific features have not been explored[21].

Here, we present CAREMIL (Cell Aggregation, Explainable, MIL), an attention-based MIL framework optimized for hematologic liquid biopsy interpretation. CAREMIL integrates both cell morphology and cytometric composition at the instance level to generate accurate, interpretable slide-level predictions. We evaluate CAREMIL across three diagnostically distinct hematologic malignancies: acute leukemia (AL), characterized by increased circulating blasts; hairy cell leukemia (HCL), marked by morphologically abnormal lymphocytes; and myelodysplastic syndromes (MDS), distinguished by subtle dysplastic (morphologic) changes across multiple hematologic lineages. To support cell-level representation learning, we leverage DeepHeme[14], a domain-specific encoder trained on hematologic cell classification, applying its feature embeddings for the first time, to the best of our knowledge, to weakly supervised, slide-level diagnosis.

CAREMIL consistently outperforms gated MIL architectures across multiple encoder types, with the greatest performance gains observed when

using in-domain foundational models trained on non-hematologic images. The model accurately identifies diagnostically relevant cell populations without requiring explicit cell-level supervision and remains robust even when upstream classifiers mislabel abnormal morphologic variants. These findings demonstrate the utility of attention-based weakly supervised learning for morphology-informed, quantitative diagnostics in hematology and highlight the potential for broader application across liquid biopsy specimens.

## Methods
### Dataset collection
We curated a dataset of 1017 WSIs of blood smears scanned at 400 × optical resolution (i.e., using a 40× objective) using a Hamamatsu S360 WSI scanner. These include 307 AL, 95 MDS, 41 HCLs, and 130 morphologically normal control cases (see Supplementary Table 3). Scanned images range from 2GB to 20GB and were scanned using a single z-plane with uniform refocusing points across each slide. Conventional computer vision methods that rely on manually defined image features for classification often have difficulty with WSIs, as they include complex biological structures (cell nuclei, cytoplasms, granules, etc.) with high variability in appearance across multiple patients and disease states (such as cancer grades or stages). To reduce the computation required for processing these gigapixel WSIs, we first extract regions of interest containing WBCs at 50 × optical resolution (downsampled from the highest 400 × resolution by a factor of 8) using a combination of deep learning and classical computer vision metrics. From these regions, we extract cell images of size 96 × 96 at 400 × magnification, cropped around the cell centers detected by a YOLOv8 model[22] (Fig. 1A). We examined a cohort of 573 patients (Supplementary Table 3). For each WSI, we extract ~2000 WBC images, and subsequently extract 1000-dimensional feature embeddings for each of them using the last-layer features of a ResNeXt-50 model trained specifically on the WBC classification task[23]. This study was conducted in accordance with the ethical standards of the institutional and national research committees and with the 1964 Helsinki Declaration and its later amendments. The research used de-identified, retrospective data and was reviewed by the Institutional Review Board (IRB) at [Memorial Sloan Kettering Cancer Center], which determined that the study qualified for exempt status (IRB Protocol #16-1591). As no identifiable private information was used, the requirement for informed consent was waived.

### Incorporating cell morphology into the predictions
Current computational approaches for the diagnosis of hematologic neoplasms involve predicting the blood differential through the summation of individual cell classifications. Trained ML classifiers use the cytometric vector as input to predict diagnosis. This approach loses morphological information of individual cells, which pathologists use when making diagnostic decisions. Moreover, it lacks cell-level interpretability, without which the final predictions are difficult for pathologists to interpret and audit, making it difficult to use in a clinical setting.

To incorporate individual cell morphology, we propose a trained encoder to first generate vector-embeddings of all the extracted cell images (Fig. 1B). Our MIL model then randomly chooses between 300 and 400 cell images and uses their cell embeddings to make a slide-level prediction. This is done to replicate the procedure adapted by a hematopathologist who observes ~300 cells to make a diagnosis. Explainability is built into the MIL aggregator to facilitate pathologists' interpretation of the model's predictions.

### CAREMIL model architecture
The CAREMIL architecture is built to ensure that we can take a varying number of cells as input and get relevant attention values for each cell embedding, while making a diagnosis prediction (see Fig. 1C. Once all the cell images are extracted, they are passed to a frozen image encoder, that creates embedding values that represent the morphological information in each cell image.

The cell embeddings are passed to different trainable networks, namely the aggregator network and the parallel network, both will reduce each cell's embedding size to the same dimension. The embedding outputs from the aggregator network are then passed to aggregator functions. We explore several aggregation functions (Supplementary Table 5). For example, Kraus et al.[24], showed that generalized mean can lead to better predictions in some computational pathology applications, whereas Carmichael et al.[25], showed that variance-based aggregation function can lead to better predictions by capturing intratumoral heterogeneity. These function outputs represent compositional information of the bloodstream, similar to the cytometric vector produced in the cytometric-based ML classifiers. The vectors are concatenated with the outputs of the parallel network before being fed into the self-attention layer. Here, the aggregation function outputs act as class tokens[26], similar to BERT[27] and Vision Transformers[28]. The class tokens from the attention layer are then fed into a multilayer perceptron classifier to predict logits and train the network using a cross-entropy loss.

### Case-level attention visualizations
To validate CAREMIL's interpretability in diagnosing hematologic malignancies, attention panel images were generated for cases of acute myeloid leukemia (AML), HCL, and MDS. For each case, we selected the top 10% of cells with the highest attention scores, a random sample of cells, and the lowest attention-scored cells. These panels allowed visual verification of the model's focus, with high-attention cells expected to show disease-relevant morphologies (e.g., blasts in AML, dysplastic myeloid cells in MDS, and abnormal lymphocytes in HCL). Randomly selected cells provide a sampling of the sample's overall composition, while low-attention cells should include morphologically irrelevant or non-diagnostic cells. This approach enables assessment of CAREMIL's attention mechanism in identifying cells most relevant to the diagnostic outcome.

### Limit of detection experiment
To assess CAREMIL's sensitivity for detecting AL in cases with low blast counts, synthetic patient cases were generated by combining cells from normal patients with varying proportions of blast cells. For each synthetic case, cells from normal PBS images were incrementally mixed with an increasing percentage of blast cells (ranging from 0 to 20%), simulating conditions of low to moderate blast presence. CAREMIL was then run on each synthetic patient case to evaluate the model's probability of predicting AML at different experimental blast levels (ref Fig. 3). Real patient cases with similar blast count distributions were also evaluated for comparison, allowing analysis of CAREMIL's performance across synthetic and real datasets.

### Attention mechanism evaluation
To evaluate how CAREMIL's attention mechanism selectively prioritizes specific cell types in different hematologic malignancies, we compared the distribution of high-attention cells across AL, MDS, and HCL cases. For each disease type, we identified the cell types based on DeepHeme cell classification, the top 10% highest attention cells as outputted by the MIL model. This allows for the calculation of the mean differential distribution across various cell types with the highest attention. These distributions were then compared to the overall cell population in normal samples to identify disease-specific enrichment patterns. This analysis provides insight into the model's diagnostic process, enabling validation of the attention mechanism's interpretability and effectiveness in distinguishing distinct hematologic conditions.

## Results
### Dataset curation
We curated a dataset of 1017 WSIs of PBS scanned at 400× magnification (40× objective), representing a diverse cohort of 573 patients evaluated by the hematopathology service. Each PBS was paired with a bone marrow biopsy and clinical pathology report, with diagnoses established by board-certified hematopathologists based on the full clinical context in the bone

marrow report. The dataset includes 307 cases of acute leukemia (AML), 95 cases of MDS, 41 cases of HCL, and 130 normal (NL) controls. Notably, control cases represent patients with normal overall reports, rather than healthy individuals, reflecting the real-world diagnostic setting. Each WSI contains hundreds of WBCs, spanning a wide range of disease burden and morphologic representations, including blast-negative AML and subtle dysplasia in MDS. To ensure independence across datasets, train, validation, and test splits were performed at the patient level.

## Improving the attention mechanism for slide-level predictions

MIL provides a weakly supervised framework for training models when only bag-level (i.e., slide-level) labels are available, a setting well-suited to pathology applications where instance-level annotations (e.g., per-cell labels) are often impractical. In MIL, each bag consists of a collection of instances, and the model must learn to aggregate information across instances to make a bag-level prediction. We compare our CAREMIL model with the Gated attention MIL.

Gated attention MIL is a common approach used in histology WSIs, where attention is applied by using a gating mechanism[17] [Ilse et al.]. Instance features undergo two separate nonlinear transformations, one with a tanh activation and one with a $\sigma$ activation. The outputs of these transformations are combined via element-wise multiplication before attention scoring. This gating mechanism allows the model to better modulate instance contributions by enhancing or suppressing different aspects of the feature representation prior to computing attention. However, aggregation remains a weighted sum across instances, and the model primarily focuses on selecting a few highly relevant instances for diagnosis.

CAREMIL builds upon the MIL paradigm by explicitly modeling both the individual morphology of instances and broader distributional properties of the instance population. Instead of aggregating raw cell embeddings through a weighted sum, CAREMIL first computes global statistical summaries (e.g., mean and variance) across the set of instance embeddings. These summary features are combined with the original cell embeddings and jointly processed through a transformer-style self-attention mechanism. This architecture enables CAREMIL to reason both about rare high-attention instances and about subtle global shifts in cell population characteristics, improving robustness and clinical interpretability.

## Model evaluation and performance

CAREMIL's performance was compared to two different approaches. The first is a set of optimized machine learning classifiers operating on the cell count alone (cytometry). To set a strong baseline, we performed hyperparameter optimization using a grid search over 6 different standard models and 227 different hyperparameter groups. Performance was compared with gated MIL attention[17]. We also tested the performance of both the CAREMIL and the Gated MIL architecture on four different image encoders (see Fig. 2. The first is DeepHeme, a domain-specific RexNeXt-50-based classifier, trained to identify 23 different cell hematopoietic cell classes found in the normal bone marrow. To create strong baselines for our work, we also make comparisons on UNI2-h[29] and Virchow2, self-supervised computational pathology foundation models[30]. Lastly, we also use a ResNeXt-50 model trained on ImageNet to understand the performance of general-purpose computer vision models not trained on histopathology. These models and their training hyperparameters are summarized in Supplementary Table 2.

Each approach was evaluated in four distinct clinical contexts, as summarized in Supplementary Table 6. The first task involved identifying acute leukemia (AL) from normal (NL) samples. AML encompasses both AML and acute lymphoblastic leukemia (ALL) and is defined by an increased percentage of blast cells in the peripheral blood or bone marrow. The second task focused on distinguishing MDS from normal samples. MDS is characterized by abnormal morphology (dysplasia) in hematopoietic cells. While MDS is typically diagnosed in patients presenting with cytopenias, the ability to detect myelodysplasia morphologically in patients with cytometrically normal blood counts may provide a low-cost method

for identifying individuals at risk of developing MDS, or who already have cytogenetically defined MDS. Earlier identification of these patients could enable interventions aimed at preventing disease progression.

The third task aimed to separate HCL from normal samples. HCL is defined by the presence of morphologically abnormal lymphocytes with "hairy" cytoplasmic projections, although these projections were rarely observed in this dataset due to slide preparation artifacts. HCL does not necessarily cause abnormal cytometry.

The fourth task involves distinguishing among AML, MDS, HCL, and NL, presenting a significantly more challenging quaternary classification problem compared to binary classification. The increased complexity stems from the need to separate four distinct classes with overlapping morphologic and cytometric features. For instance, AML and MDS may share dysplastic characteristics and overlapping blast counts, while all four classes can exhibit normal cytometric profiles under certain conditions. This task is further complicated by the requirement for the classifier to generalize across all classes while minimizing errors specific to each, highlighting the difficulty of achieving robust and accurate multi-class classification.

Slides were labeled with diagnoses derived from pathology reports at the case level. Model performance was evaluated based on its ability to correctly predict these case-level diagnoses. For example, a diagnosis of acute leukemia (AL) did not require detectable disease in the peripheral blood, as the diagnosis may have been based on findings from the bone marrow. This approach ensured the inclusion of cases with low-level circulating disease that might be identified by the AI system, even when undetectable by automated hematology analyzers or human pathologists.

CAREMIL outperforms Gated MIL on all tasks (see Supplementary Table 8) and performs better irrespective of the image encoder being used (see Fig. 2A). Our results also show that the hematology-specific image encoder, DeepHeme, demonstrated substantially better performance compared to large general-purpose pathology foundation models like UNI2-h or Virchow2. Both the encoder and the aggregator model architecture play an important role in the overall performance. The CAREMIL model architecture with the DeepHeme image encoder demonstrates superior performance compared to other models and encoders for the hematology domain.

Supplementary Table 8 summarizes the average AUROC scores and macro-averaged F1-scores for each model and experiment. In the MDS vs NL and multiclass task, UNI2-h achieved slightly higher performance, though the difference was not significant. Across all four experiments, models using the CAREMIL architecture performed significantly better than their Gated MIL equivalent. For further baseline experiments, we also trained the datasets on DS-MIL[31], a commonly used pathology aggregation model. However, it is designed with an MIL approach where a bag is considered to have a positive label if even a single positive instance is found. This makes it improper for our application since a single blast cell will not change the diagnosis of the patient to AML. Additionally, cytometry-based (e.g., based on cell counts alone) machine learning classifiers consistently outperformed MIL models that relied on ImageNet encoders.

## Explainability and patient safety

AI models deployed in medical environments must include a layer of explainability to ensure patient safety and clinical utility. A key aspect of explainability is enabling physicians to rapidly determine whether the model's performance is based on sound biology and not spurious correlations. Additionally, adding explainability outputs can facilitate broader analyses on how abnormal cells influence diagnostic decisions. For this reason, we developed a workflow designed to provide pathologists with interpretable outputs for each sample. The output consists of a figure showcasing three key cell subsets (see Supplementary Fig. 7): the cells with the highest attention values (top row), a random selection of cells to represent the whole-slide context (middle row), and the cells with the lowest attention values (bottom row). This structure allows pathologists to confirm whether the model is focusing on diagnostically relevant cells while avoiding irrelevant or morphologically normal cells.

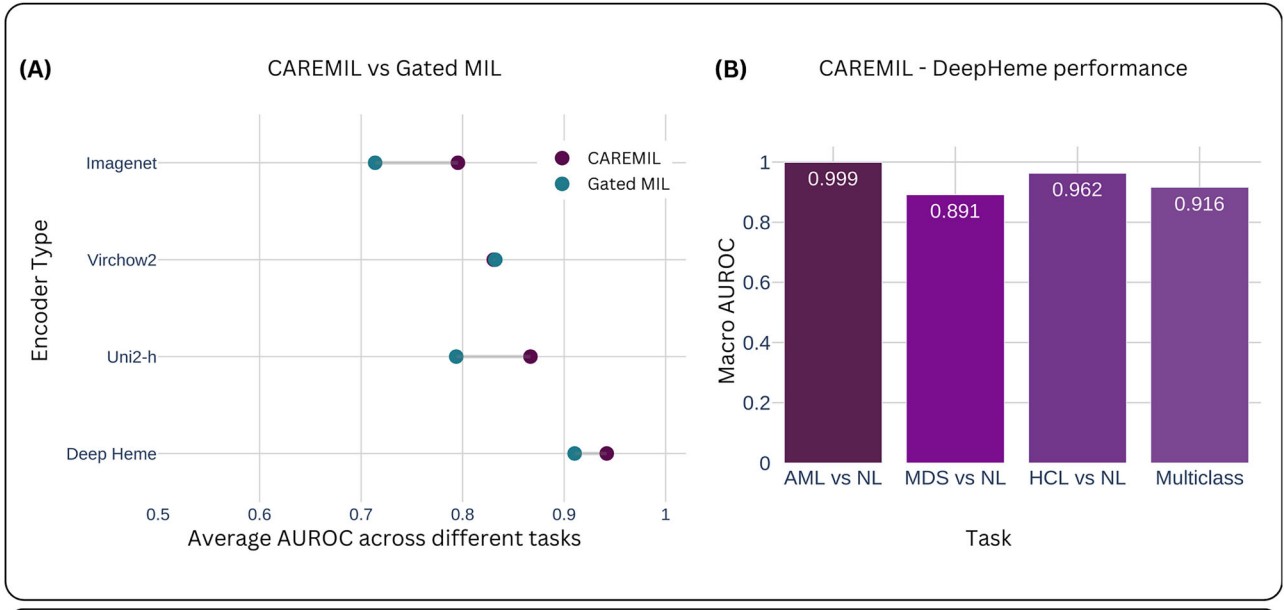

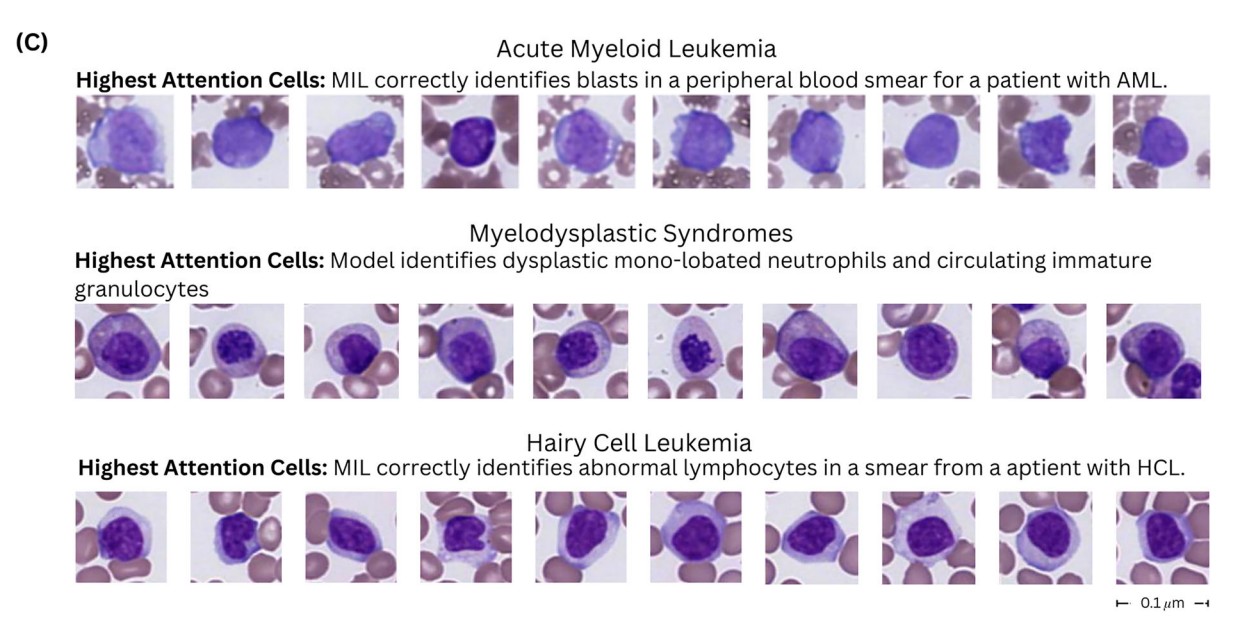

**Fig. 2 | Performance and explainability of CAREMIL across encoder types.**
**A** Comparison of CAREMIL and Gated MIL performance across different encoder types. The plot shows the average AUROC across multiple diagnostic tasks using four different pretrained encoder types: ImageNet, Virchow2, Uni2-h, and DeepHeme. Each horizontal pair of dots represents the performance of CAREMIL (purple) and Gated MIL (teal) using the same encoder. CAREMIL consistently outperforms or matches Gated MIL across all encoder types, with the largest performance margin observed for the Virchow2 and Uni2-h encoders. Horizontal lines connect paired values for visual comparison. **B** CAREMIL + DeepHeme is the best performing combination of architecture and encoder and its individual performance on the 3 binary classifications and 1 multiclass classification is shown. **C** CAREMIL provides explainable diagnostic decisions by highlighting the cells that drive each prediction. The top 10 highest attention cells generated by CAREMIL + DeepHeme are displayed from selected examples from AML, MDS, and HCL, respectively.

The top row specifically highlights the morphology of disease cells, offering a clear visual confirmation of abnormal cells. In cases of very low disease burden, where abnormal cells may not appear in the random selection, the top row ensures the clinician can still validate the diagnosis. The middle row provides a representative sampling of randomly selected cells, enabling clinicians to quickly estimate the disease burden in the patient by assessing the proportion of abnormal cells. The bottom row allows clinicians to verify that the model correctly identifies and de-emphasizes cell types that are least diagnostically relevant, ensuring it focuses on cells that contribute meaningfully to the predicted diagnosis. While the figures presented in this study are limited to 10 cells per row for visual clarity, a clinical deployment would include a larger number of cells. This would make it possible to accurately estimate disease burden in cases with a low percentage of abnormal cells.

In the context of model explainability, specific patterns of cell morphology were observed for each disease state, corresponding to the high attention cells. Figure 2C is composed of disease-specific examples of high attention-based outputs that demonstrate the model's capacity to align its predictions with clinically relevant morphological features while providing a broader context for disease burden and cell-type distribution.

In Fig. 2C, the top panel shows that in a sample with AML, the high-attention cells consistently included leukemic blasts characterized by fine chromatin and high nuclear-to-cytoplasmic ratios. The next panel shows a case of MDS, with high-attention cells including examples of both dysplastic

**(A)**

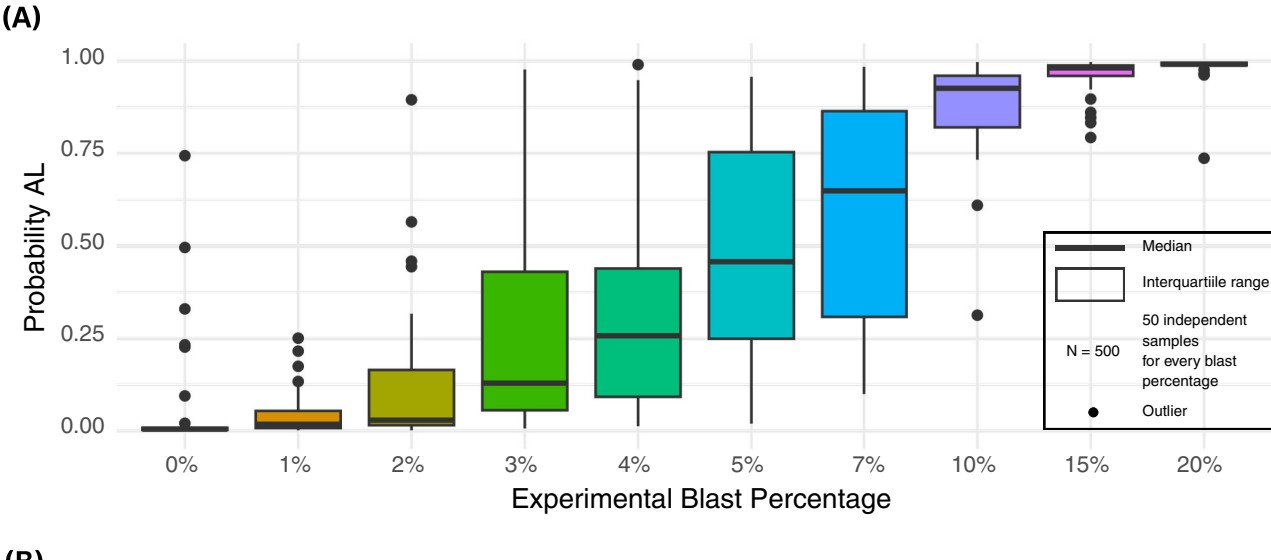

**(B)**

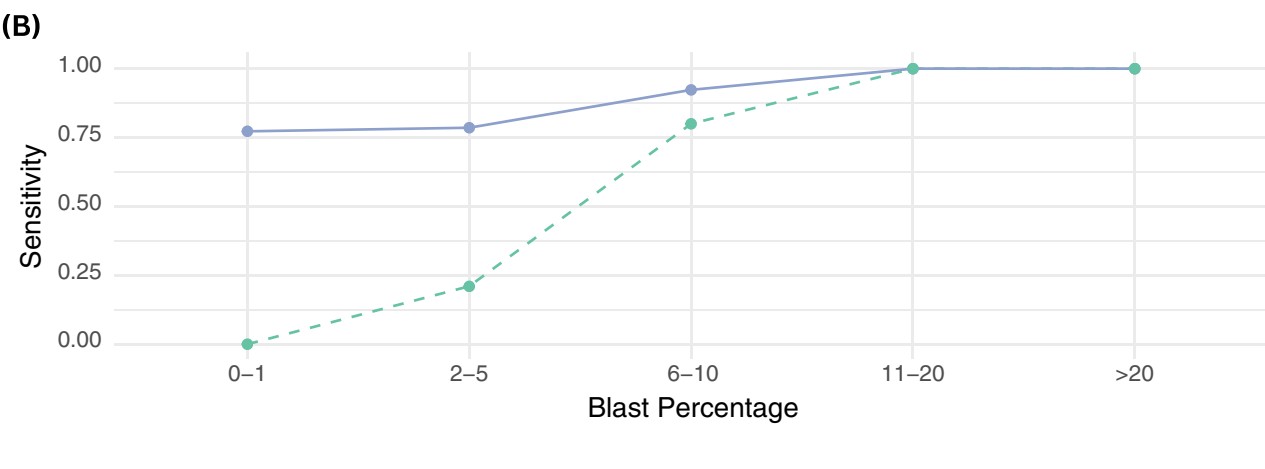

**Fig. 3 | CAREMIL ablation studies. A** Limit of detection study. This figure illustrates the CAREMIL scores across a range of blast percentages in synthetic patients. Synthetic patients were generated by combining a specific percentage of randomly selected blast cells from acute leukemia patient samples into random cells taken from normal patient samples. **B** CAREMIL performance in minimal leukemia detection. This graph compares the sensitivity of CAREMIL for diagnosing leukemia in real patients (solid line) versus synthetic patients (dashed line) across various blast percentage categories. Sensitivity increases with higher blast percentages in both patient groups. However, CAREMIL performs better on real patients, suggesting the model is incorporating data from non-blast cells into its prediction. Surprisingly, it achieves a sensitivity of 75% for samples with blast count  < = 1%, which might be indistinguishable from normal samples by traditional methods.

neutrophils and immature granulocytes, which, when circulating in peripheral blood, are characteristic of MDS. The bottom panel shows an HCL sample, where the high-attention cells highlight the characteristic abnormal "hairy cell" lymphocytes associated with the disease.

Supplementary Fig. 7 highlights an additional case of MDS. High-attention cells in this case include dysplastic hypogranular neutrophils, monolobated neutrophils, dysplastic erythroblasts exhibiting nuclear-cytoplasmic dyssynchrony, and abnormal immature granulocytes circulating in the peripheral blood. The inclusion of two examples for MDS demonstrates the model's ability to identify different types of dysplasia across specimens, reflecting the diversity of morphological abnormalities seen in MDS, which represents a family of syndromes, rather than one specific entity. These findings underscore the model's adaptability in detecting patient-specific manifestations of dysplasia, providing critical information for diagnosis and model explainability.

### Limits of detection in synthetic patients

To evaluate the sensitivity of CAREMIL across varying blast percentages, we generated a dataset of synthetic patients by combining cells from normal patients with varying proportions of blast cells. Blast cells from all 307 AML patients were collected, and normal cells from all 130 normal PBS images

were collected. For each synthetic case, cells from the normal cells group were incrementally mixed with an increasing percentage of cells from the blast cells group (ranging from 0 to 20%). 50 synthetic patients were generated for every blast cell mixture ratio, simulating conditions of low to moderate blast presence. For every simulated patient, we randomly choose between 300 and 400 cells to pass to CAREMIL to make a prediction. This is done to replicate the process during training. Leukemia is typically diagnosed when blast percentages exceed 20%. However, lower blast levels may be observed in post-treatment patients, cases where the disease is predominantly confined to the marrow, or for certain disease genotypes. As illustrated in Fig. 3A, the probability of correctly diagnosing AL increases with rising blast percentages. Using a cutoff of 0.5, at 0% blast proportion, the algorithm identifies all but one sample as normal, and at 20%, the algorithm identifies all samples as AL.

### False positive review reveals true positive relapse

Interestingly, CAREMIL's performance at low levels of circulating blasts in synthetic datasets appeared substantially lower than in the WSI dataset. The blast percentages in our WSI dataset span from 0 to 100% in cases of AL. Slides were included in the study based on pathology reports diagnosing AL, even if circulating blasts were not detected by hematology analyzers or visual

inspection noted in the pathology report. In cases where circulating blast percentages were below 20%, patients typically exhibited higher blast counts in the bone marrow, consistent with the diagnostic criteria for AL.

In Fig. 3B, the continuous line delineates the performance of CAR-EMIL across varying levels of circulating disease. For clarity, we categorized our test data into different bins: high (> = 20%), low (<20%), very low (< = 10%), extremely low (< = 5%), and invisible leukemia (< = 1%). CAREMIL demonstrates high performance at low levels of blasts on real patients compared to synthetic patients, which are represented by the dashed line. Surprisingly, CAREMIL achieves 75% sensitivity, even in cases with 0% blast count. This suggests that CAREMIL may be recognizing changes in the composition or morphology of the non-blast cell population. However, these changes are not necessarily specific to AL and could also occur in other hematological disorders. For instance, they may result from marrow damage caused by leukemia cells, leading to alterations in peripheral blood morphology or cytometry. Alternatively, AL might induce specific but subtle, multifactorial changes in non-blast cells that are not easily recognizable by pathologists but identifiable by CAREMIL.

Although missed diagnoses in cases with low-level diseases are anticipated, encountering false positives where normal slides are misclassified as AL is less expected. In our test set, no such instances occurred. However, in the validation set, four cases were identified. Upon review by our pathologist colleagues, two were found in patients with a history of leukemia but no current detectable disease. Another involved a patient who had recently received granulocyte colony-stimulating factor (G-CSF) treatment prior to a bone marrow transplant for lymphoma. For patients whose bone marrow stem cells are being harvested for transplant, G-CSF treatment is performed to massively increase circulating hematopoietic stem cells (blasts) prior to harvesting.

The most notable "false positive" involved a patient with a history of AML in remission who presented to the clinic with an enlarged lymph node. Although imaging raised concerns for relapse, initial diagnostic evaluations, including a PBS, bone marrow biopsy, bone marrow flow cytometry, and bone marrow biopsy genetic testing for a known *FLT3* mutation, did not confirm disease recurrence. Based on this workup, the bone marrow biopsy report was classified as morphologically normal, resulting in its inclusion in our test set as a morphologically normal sample. However, a lymph node biopsy confirmed AML, prompting a repeat bone marrow biopsy three weeks later, which also revealed AML with an 80% blast count. This suggests that CAREMIL may have identified the presence of AML approximately three weeks earlier than the standard diagnostic workup.

### Model-derived morphometric signatures reflect underlying disease biology

The attention values created from the model provide valuable interpretability beyond the individual patient level. For each individual patient, we create an attention panel as shown in Fig. 3, through which a pathologist can verify the cells being looked into for each prediction. To evaluate the overall effectiveness of our interpretability mechanism beyond individual attention maps, we compare the cell type distribution within the top 10% most-attended cells in each disease category to the distribution of all cells in the same disease category. This highlights the cell types that are relevant to the decision-making for each disease category.

As shown in Fig. 4A, the relative composition of blast cells in the top 10% highest attention cells is 2.5 times higher than in the total population, which indicates that the attention mechanism is correctly identifying blasts as the important cell class for diagnosing AML. Supplemental Fig. 4C extends this analysis using a log-odds ratio to highlight the contribution of rare cell classes to the disease signature. Notably, we observe increases in mitotic bodies, orthochromic/polychromatophilic erythroblasts, plasma cells, and basophils. While mitotic bodies are rarely seen in circulation, their presence is elevated in AL, and the model appears to have captured this pattern. The increases in other rare circulating cell types likely reflect underlying marrow stress, further supporting the model's ability to detect subtle morphological indicators of disease.

In Fig. 4B, the MDS disease signature is illustrated. For MDS patients, the model assigns high attention to lymphocytes, blasts, and basophils. We hypothesize that the increased attention to lymphocytes is due to the relative lymphocytosis observed in patients with myeloid cytopenias, a hallmark feature of MDS. On further examination, some patients in the cohort had lymphocyte percentages in excess of 50% of all circulating WBCs. Likewise, abnormal blasts and basophilia are also commonly seen in MDS.

The comparative attention histogram for HCL shows that attention is less focused on lymphocytes. Thus, the DeepHeme model does not classify the hairy cells as normal lymphocytes. Fig. 4C shows that the high attention category contains overrepresentation of monocytes and blasts. Pathologist review revealed that the cells interpreted as blasts and monocytes by DeepHeme in these samples were hairy cells. DeepHeme is trained on patients with no known neoplastic disease. Thus, hairy cells were absent from the training corpora. Despite this limitation, the CAREMIL model determined that the features associated with hairy cells were important to the underlying diagnosis, despite the encoder model (DeepHeme) not being trained to specifically classify hairy cells. This highlights the strength of the CAREMIL model, which does not rely on the cell-level classification provided by the encoder. Even when the cell labels are incorrect, the model can learn to identify the important cell morphologies provided by the feature embedding from the encoder.

### Model attention aligns with diagnostic cell identity

The disease signature helps pathologists review the cells that the model takes into consideration for diagnosis, along with the weight given to each cell. However, this pictorial and manual verification is not enough to prove the model's capabilities. We want to understand the overlap between the cells that are given a high attention value (top 25% by attention value) by the model, vs the diagnostically relevant cells identified for that patient. We choose a sample that has been diagnosed with AML for this analysis, as AML has a simple biomarker of the presence of blast cells. In Fig. 5, almost all high attention cells are blast cells, showcasing the practical utility of the model to identify important diagnostic markers without the need to explicitly teach the model-specific biomarkers for every disease.

### Conclusions

Overall, we proposed a framework, CAREMIL with DeepHeme, which is the best performative model for hematological diagnostic tasks. Instead of traditional diagnosis of hematologic cancers, which rely on enumerating different blood cell types (the blood differential) and applying heuristic rules alongside morphological analysis of individual cells, our pipeline leverages a hematologic domain-specific encoder to integrate cellular morphology with cytometric features for case-level diagnosis of hematologic malignancies from blood smears.

Unlike prior AI efforts that classify cells individually and base diagnosis on cell differentials alone, our model also integrates cell-level morphologies across the entire case to inform diagnostic predictions. Moreover, conventional cell classifiers often offer limited interpretability, making it difficult to meaningfully involve pathologists in the diagnostic process[32], a limitation CAREMIL explicitly addresses by providing cell-level attention scores.

In computational pathology for tissue sections, WSIs are analyzed through a tiling approach for slide-level diagnosis. However, these methods are unsuitable for hematology, where diagnostically relevant cells, such as WBCs, are sparsely and randomly distributed. Our proposed methods overcome these limitations by implementing a cell-aggregation-based MIL method with attention mechanisms that prioritize diagnostically important cells[33] for pathologist review, facilitating accurate slide-level predictions.

Our findings demonstrate that CAREMIL, paired with DeepHeme cell image encoding, achieves substantial improvements in diagnostic accuracy by combining individual cell-level morphological features with cytometric composition, which is pivotal for diagnosing and staging hematologic malignancies. Notably, our pipeline achieved AUROC scores of 0.999, 0.891, and 0.945 for AML, MDS, and HCL, respectively, underscoring its

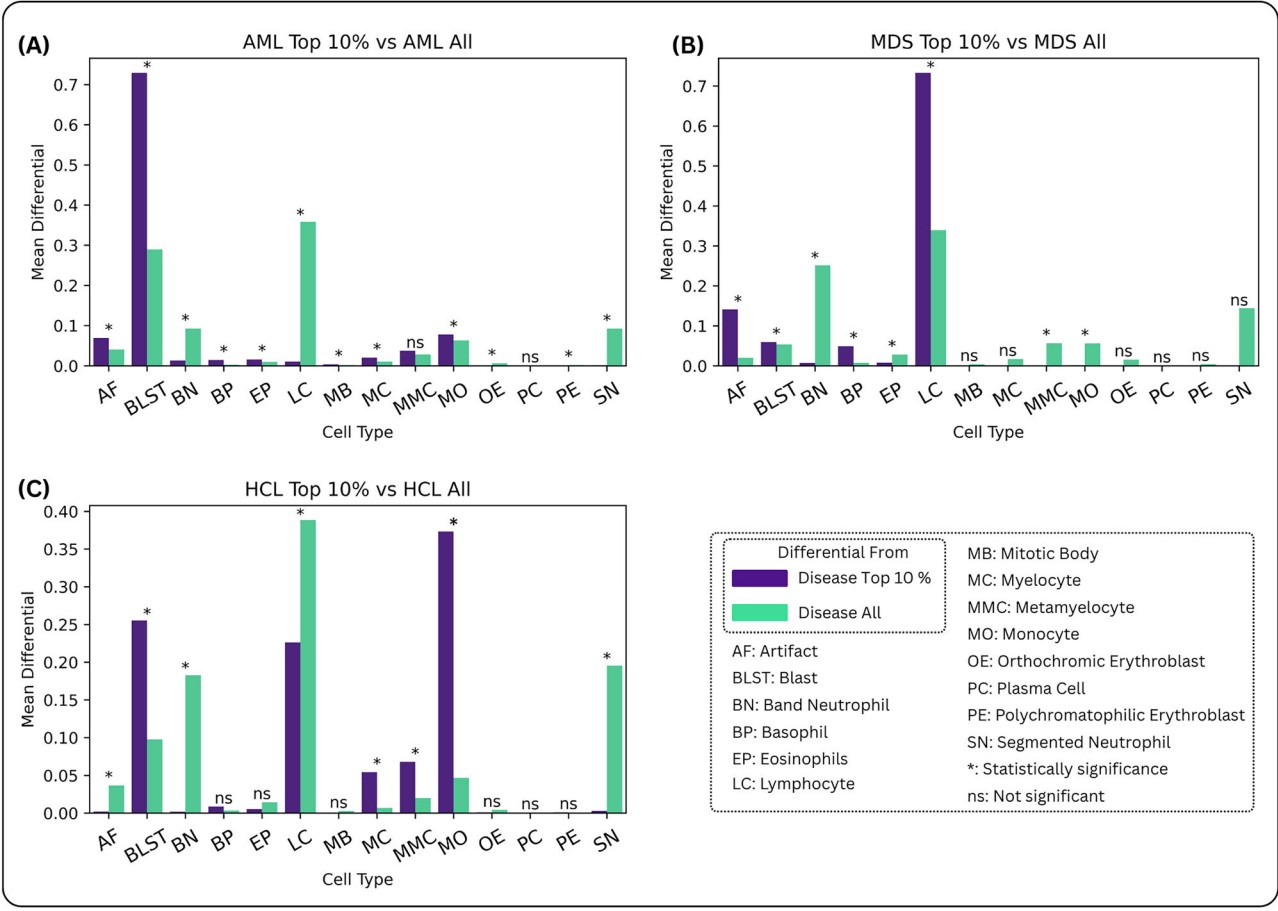

**Fig. 4 | Comparative attention histograms.** Each graph shows the distribution of cell types among the top 10% highest-attention cells, identified by the CAR-EMIL + DeepHeme architecture, in correctly-classified cases, compared to the overall distribution of cell types in the same cases. A chi-squared test is used to determine the statistical significance of the enrichment for each cell type in the top 10% highest-attention cells vs. all cells. A Bonferroni correction is used to account for the multiple hypotheses being tested. Statistically significant differences are highlighted with *. Bar chart for **A** AML, **B** MDS, **C** HCL.

**Fig. 5 | Verifying high attention cells with diagnostically relevant cells.** This is a UMAP of the cell embeddings of a patient diagnosed with AML. High attention cells are shown in red, with low attention cells in blue. Blasts, a biomarker of acute leukemia, are depicted as filled diamonds, whereas other cells are empty circles. Most high attention cells (top 25% cells by attention value from the model) are blast cells, which are a biomarker for AML diagnoses (filled red diamonds).

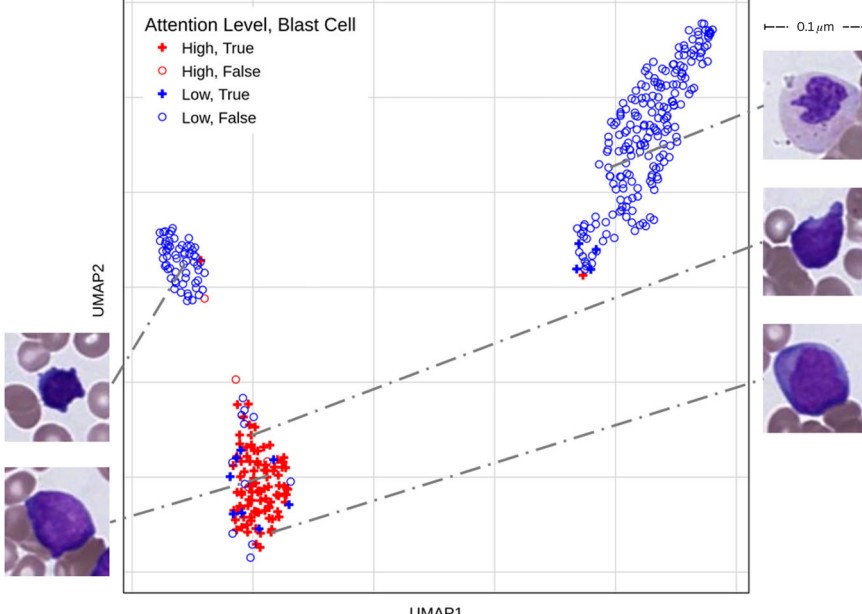

clinical relevance across diverse disease contexts. We showed that the use of DeepHeme outperforms other image encoders, including pathology foundation models and ImageNet-trained encoders. This result highlights the importance of domain-specific feature extraction for arriving at the most accurate diagnosis.

An important aspect of the MIL strategy for hematologic diagnosis is the explainability framework we highlight. By employing an attention-based MIL aggregator, the model identifies cells critical to the diagnostic prediction, allowing pathologists to validate and interpret the model's rationale in the context of each case. This explainability, alongside performance, is essential for clinical adoption, as it enables medical professionals to ensure model predictions align with established diagnostic criteria, thereby promoting clinician trust[34] and patient safety.

The attention-based weakly supervised approach employed in CAREMIL offers unique insights into underlying disease processes by identifying diagnostically relevant cells, rather than relying solely on quantitative metrics like blast counts. Specific genotypes of AML associated with aggressive phenotypes do not require a specific blast percentage for diagnosis, according to the 5th edition of the World Health Organization Classification of Haematolymphoid Tumors[35]. This pipeline can be replicated with different disease labels to build an automated biomarker identification system that will emerge as a valuable tool in the future to stratify high-risk patients, enabling earlier and more precise interventions. This approach can also be used to develop outcome prediction models that are morphology-informed, using outcome labels rather than diagnosis labels.

The ability to assign attention values to individual cells based on their relevance to the diagnostic outcome allows for a nuanced interpretation of WSI data that captures both explicit and latent features across the entire sample. This model of disease characterization holds promise for future applications, where attention-based diagnostics may surpass traditional metrics in robustness and specificity, especially for diseases with complex presentations. By focusing on pathologically significant cells rather than fixed thresholds, CAREMIL could enable more personalized and accurate diagnostics that reflect the continuum of disease rather than categorical criteria.

While our techniques demonstrate strong diagnostic potential, several limitations underscore areas for future development. The current approach does not support end-to-end training or fine-tuning, which may be necessary for applications lacking pre-existing in-domain feature encoders (non-hematopoietic cells in cytology applications, for example). Additionally, DeepHeme generates feature embeddings from a relatively modest-sized ResNeXt-50 model, and future implementations may benefit from larger, self-supervised architectures to achieve even better performance, especially as diagnostic complexity increases.[36]

Another consideration is the diagnostic spectrum of the cell-based MIL strategy. This study addresses a four-way classification, yet the full range of hematologic diagnoses is far more extensive. Expanding the study to encompass additional conditions will require large, diverse datasets and further testing to ensure reliability across a broader range of hematologic pathologies. Moreover, while this study focuses solely on PBSs–a practical choice for screening due to their low cost and minimally invasive nature—a more accurate diagnosis would be achieved by integrating data from bone marrow aspirates, core biopsies, immunohistochemistry, flow cytometry, clinical text histories, and genomic data. Conveniently, the current pipeline provides an encoding strategy that could allow for incorporation in a multimodal study that includes these diverse data types.

The MIL architecture, CAREMIL, along with the DeepHeme cell encoder, represents a noteworthy advancement in computational hematology, providing an accurate, interpretable framework for diagnosing hematologic diseases using WSIs of PBSs. Leveraging a cell-aggregation-based MIL approach not only preserves essential morphologic information of WBCs but also achieves superior diagnostic accuracy compared to cytometry-based models. Future work should explore expanding this pipeline's diagnostic scope and integrating multimodal data to enhance its clinical utility in hematology.

## Data availability

For follow up instructions please contact Chad Vanderbilt (vanderbc@mskcc.org) or Gregory Goldgof (goldgofg@mskcc.org) for initiating a data transfer agreement (DTA) with MSKCC Legal Dept. The DTA process will take 6–9 months, and is at the discretion of the legal department. No protected health information will be shared under any circumstances. The source data for Fig. 3 is in Supplementary Data 1. The source data for Fig. 4 is in Supplementary Data 2.

## Code availability

The code for CAREMIL is available on GitHub and has been archived on Zenodo[37].

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

## Acknowledgements

This research was supported by funding from the Cancer Center Support Grant from the NIH/NCI (P30CA008748), the Warren Alpert Foundation through the Warren Alpert Center for Digital and Computational Pathology at Memorial Sloan Kettering Cancer Center, and the MSK Technology Development Fund. We extend our gratitude to the MSK Hematopathology and Digital Pathology Services for their contributions.

## Author contributions

Conceptualization: Si.S., G.M.G., and C.V. Methodology: Si.S., Sh.S., Z.Y., R.G., D.C.W., N.K., S.N., N.S., J.G.V.C., B.F., E.S.Y., Ar.S., C.C.J., I.C., G.M.G., and C.V. Experiments: Si.S., Sh.S., G.M.G., and C.V. Data curation and acquisition: G.M.G., I.S.I., K.H.B., J.B., L.W., I.S.I., M.Z., L.B., M.F., M.Y., W.X., La.M., M.R., A.D., and O.A. Manuscript writing: Si.S., Sh.S., I.C., G.M.G., and C.V. Funding acquisition: G.M.G., C.V., and O.A. Reviewal and approval of manuscript: Si.S., Sh.S., Z.Y., R.G., D.C.W., K.H.B., D.D., L.W., N.K., S.N., N.S., J.G.V.C., B.F., S.P., E.S.Y., A.K., A.S., A.M., J.B., I.S.I., C.C.J., An.C., L.B., D.K., B.K., M.F., Al.C., M.Y., S.I.M., M.Z., S.M., O.A., La.M., W.X., M.R., O.L., A.D., I.C., C.V., and G.M.G

## Competing interests

L.B. reports stock ownership in Exact Sciences. M.Y. reports consulting for Janssen Research and Development. S.M. reports equity interest in Daboia Consulting LLC and professional services for Janssen Pharmaceuticals, Medical Case Management Group, North American Thrombosis Forum, and Physicians' Education Resource. W.X. reports research support from Stemline Therapeutics. M.R. reports serving as a scientific advisory board member with equity support at Auron Pharmaceutical, research funding from Celularity, Roche-Genentech, Beat AML, and NGM, and travel funding from BD Biosciences. O.L. reports consulting fees from Janssen Biotech and Hologic, and support for professional activities from the American Society of Cytopathology. A.D. reports consulting for Seattle Genetics, Takeda, EUSA Pharma, AbbVie, Peerview, Physicians' Education Resource, Incyte, and Loxo, as well as research support from Roche and Takeda. C.V. reports equity interest and intellectual property rights in Paige.AI, Inc., and a consulting and advisory role for Paige.AI, Inc. G.M.G. reports equity interest in HemeAI, Inc. The following authors declare no competing interests: Si.S., Sh.S., Z.Y., R.G., D.W., K.B., D.D., L.W., N.K., S.N., N.S., J.C., B.F., S.P., E.Y., A.K., Ar.S., A.M., J.B., I.I., C.C.J., Al.C., Ai.S., D.K., B.K., M.F., An.C., S.M., M.Z., O.A., La.M., and I.C.

## Additional information

**Siddharth Singi** [1] ✉, **Shenghuan Sun** [2], **Zhanghan Yin**[1,3], **Riya Gupta**[1], **Dylan C. Webb**[1,3], **Khawaja H. Bilal**[1], **Deepika Dilip**[4], **Linlin Wang**[5], **Neeraj Kumar**[1], **Swaraj Nanda**[1], **Nicolas Sanchez**[1,3], **Jacob G. Van Cleave**[1], **Brenda Fried** [1], **Sean Paulsen** [1], **Ethan S. Yan** [1], **Ali Kamali**[1], **Argho Sarkar**[1], **Allyne Manzo** [1], **Jeeyeon Baik**[1], **Irem S. Isgor**[1], **Cesar Colorado-Jimenez**[1], **Anthony Cardillo**[1], **Leonardo Boiocchi**[1], **Aijazuddin Syed**[1], **David Kim** [1], **Brie Kezlarian-Sachs**[1], **Maly Fenelus** [1], **Alexander Chan** [1], **Mariko Yabe**[1], **Samuel I. McCash** [1], **Menglei Zhu**[1], **Simon Mantha** [6], **Orly Ardon** [1], **Lauren McVoy**[1], **Wenbin Xiao** [1], **Mikhail Roshal**[1], **Oscar Lin**[1], **Ahmet Dogan** [1], **Iain Carmichael**[7], **Chad Vanderbilt** [1,10] & **Gregory M. Goldgof** [1,8,9,10]

[1]Department of Pathology and Laboratory Medicine, Memorial Sloan Kettering Cancer Center, New York, NY, USA. [2]Bakar Computational Health Sciences Institute, University of California, San Francisco, CA, USA. [3]Department of Statistics, University of California, Berkeley, CA, USA. [4]New York Medical College, Valhalla, NY, USA. [5]Department of Laboratory Medicine, University of California, San Francisco, CA, USA. [6]Department of Medicine, Memorial Sloan Kettering Cancer Center, New York, NY, USA. [7]School of Data Science and Society; Department of Pathology and Laboratory Medicine, University of North Carolina-Chapel Hill, Chapel Hill, NC, USA. [8]Department of Pathology and Laboratory Medicine, Weill Cornell School of Medicine, New York, NY, USA. [9]Halvorsen Center for Computational Oncology, Memorial Sloan Kettering Cancer Center, New York, NY, USA. [10]These authors jointly supervised this work: Chad Vanderbilt, Gregory M. Goldgof ✉e-mail: singis@mskcc.org

