## [Transparent Peer Review file · Communications Medicine]

Interpretable Multiple Instance Learning for Hematologic Diagnosis from Peripheral Blood Smears

Corresponding Author: Mr Siddharth Singi

Version 0:

Reviewer comments:

Reviewer #1

(Remarks to the Author)
Brief Summary

The manuscript presents a deep learning pipeline for disease diagnosis (AML, MDS, and HCL versus normal) from whole slide images of peripheral blood. The proposed pipeline integrates a domain-specific hematology encoder with a Multiple Instance Learning (MIL) framework. This approach achieves high classification performance and offers interpretability by identifying the specific cells driving the diagnosis- highlighting its potential for clinical application.

Overall Impression

The manuscript is well-written, with clear explanations of the methodology and a strong justification for the clinical importance of the work. I recommend that this manuscript be Accepted with Minor Revisions.

Strengths

The manuscript's primary strength lies in its thorough and robust analysis. Specific points of merit include:

Rigorous Evaluation: The authors rigorously evaluate the CAREMIL pipeline by testing its performance with various encoders (domain-specific, pathology foundation, and general-purpose) and against multiple cytometry-based machine learning classifiers with hyperparameter tuning.

Sound Validation: The validation methodology is sound- employing a patient-level split for the training, validation, and test sets to prevent data leakage.

Model Sensitivity Analysis: The analysis of the model's limit of detection using synthetically generated samples is a valuable contribution, providing insight into the model's sensitivity to the percentage of blast cells.

Effective Visualizations: The visualizations are particularly effective. The figures comparing the highest-attention, randomly selected, and lowest-attention cells compellingly demonstrate both the model's performance and its potential for clinical interpretability.

Methodological Transparency: The authors provide sufficient detail regarding data collection, processing and evaluation methodologies, allowing for a clear understanding of the work.

Suggested Improvements

Clarity on Synthetic Dataset: The manuscript would benefit from additional details about the synthetic dataset. Specifically, clarifying the total size of this dataset and the sampling strategy for blast and normal cells (e.g., whether cells for a single synthetic sample were drawn from one or multiple patients) would be helpful.

Details on the CAREMIL Model and Comparison to State-of-the-Art: The code repository was not accessible at the time of

review. Providing access or including more architectural details on how CAREMIL improves upon the standard gated MIL formulation would strengthen the paper. Furthermore, to better position their work within the current literature, a comparison or discussion against recent attention-based MIL methods, such as SCEMILA (PLOS Digital Health, 2023), would be valuable.

Discussion of Classifier Performance: Supplementary Table 5 presents several interesting results that warrant further discussion in the main text. CAREMIL's superior performance over gated MIL is clear, as is the benefit of the domain-specific encoder. Notably, the high accuracy achieved by the simpler cytometry-based ML classifiers is also striking. While the interpretability of MIL is a clear advantage, the manuscript would be more compelling if it discussed potential scenarios or specific cases where these simpler classifiers might fail, but the MIL-based approach would succeed. To supplement the AUC ROC results, please add Precision-Recall curves to allow comparison of model performance in terms of precision versus yield, which is critical for understanding the real-world impact on false positives and false negatives.

Reviewer #2

(Remarks to the Author)

Summary of Manuscript

The manuscript introduces CAREMIL (Cell AggRegation, Explainable, Multiple Instance Learning), an attention-based MIL framework designed for slide-level diagnosis of hematologic malignancies from peripheral blood smears (PBSs). Unlike conventional approaches that either focus on single-cell classification or cytometric profiles, CAREMIL integrates morphological features from individual white blood cells with broader cell population statistics. The authors pair CAREMIL with DeepHeme, a domain-specific cell image encoder, and compare performance across multiple encoders (DeepHeme, UNI2-h, Virchow2, ImageNet). The method is evaluated on 1,017 PBS WSIs covering AML, MDS, HCL, and normal controls. CAREMIL consistently outperforms Gated MIL and cytometry-based baselines across various tasks. The framework includes an explainability component that highlights diagnostically relevant cells, aiding clinical validation. Additional analyses explore detection limits, false positive review, and disease-specific morphometric signatures.

General Comments

This is an important contribution to quantitative hematology. Modeling PBS diagnosis directly at the slide level, jointly leveraging cell morphology and composition, addresses a real clinical gap and moves beyond simple "sum of cell types." The interpretability elements (attention-based cell panels, disease-specific signatures) are clinically appealing. However, to meet translational standards, the manuscript requires substantial strengthening on clarification and reproducibility.

Strengths

- + Clinically aligned task
- + Strong results
- + Useful interpretability

Weaknesses

- Limited baseline diversity
- Reproducibility is in question

Major issues

1. Slide-level labels are derived from bone marrow biopsy results and clinical pathology reports, whereas the data used are WSIs of peripheral blood smears. While this labeling strategy reflects clinical practice, it can project a case-level positive label onto a PBS that may show little or no peripheral evidence, especially in cases with low blast counts. The overall experimental design would be more strongly justified if the model could provide supporting evidence that PBS morphology or composition differs systematically between normal and abnormal samples, even in such challenging scenarios.
2. The manuscript and supplementary materials alternate between describing the task as AL vs NL, while also clarifying that AL = AML + ALL. Please ensure that the terminology and experiments are consistent, either use AL (AML + ALL) throughout or restrict the analysis explicitly to AML only.
3. The reproducibility of the study is limited by both clinical and commercial constraints. All experiments were conducted on a single in-house dataset using privately owned foundation models. Although a GitHub link is provided, it is currently not functional or usable. To strengthen the generalizability and reproducibility of the work, it would be highly valuable to include validation on at least one external dataset.
4. I recommend including a broader set of MIL method comparisons, since the primary novelty of this work lies in the introduction of statistical aggregation tokens. At present, most baselines focus on variations of the backbone feature encoder, while the gated MIL variants perform worse than even the conventional cytometry counting-based models. Adding stronger or more diverse MIL baselines would better contextualize the contribution of the proposed approach.
5. What distinguishes the high-attention samples identified by CAREMIL from those highlighted by gated MIL? In particular, does CAREMIL demonstrate an improved ability to extract clinically relevant cells, suggesting that its attention mechanism is better aligned with meaningful morphological features?
6. Please provide additional model specifications, including the number of parameters and the floating-point operation counts (FLOPs), as model scale can have a substantial impact on performance. Also, please clarify the backbone architecture used across all models, are they all based on ResNeXt-50, or do some employ different encoders?
7. In the proposed design, the statistical tokens are combined with the cell image tokens before entering the self-attention layer. Consequently, the output is reweighted with respect to all cells. Could you clarify what the attention weights assigned to the statistical tokens look like? If the attention primarily emphasizes the individual cell tokens rather than the statistical tokens, the approach may effectively reduce to a conventional MIL formulation, diminishing the distinct contribution of the

statistical tokens.

Minor issues

1. Several figures and tables in the supplementary material do not follow the order in which they are cited in the manuscript, which makes cross-referencing difficult.
2. The caption for Supplementary Figure 10 appears to be incomplete and should be revised for clarity.

Typos

page 4 row 100: where attention is is applied
page 6 row 120: RexNeXt-50-based classifier
methods supplementary Section 1.2: vector-embeddings
figure supplementary 4: from the the two differential types

Version 1:

Reviewer comments:

Reviewer #2

(Remarks to the Author)

The authors have adequately addressed my previous concerns by clarifying the dataset construction, providing additional methodological details, strengthening model comparisons, improving performance reporting, and resolving terminology and reproducibility issues. While some aspects (such as external validation and more quantitative assessment of attention quality) could still be further enhanced in future work, the revision substantially improves the clarity and robustness of the manuscript, and my earlier concerns are sufficiently addressed.

A minor note: for completeness, please remember to include the caption for Supplementary Figure 10 in the supplementary materials.

Revision commentary:

Respected editors at Nature Communications Medicine,

Thank you so much for accepting to review our paper and taking the time to get back to us with detailed and valuable feedback to improve the clarity and impact of our paper. Please see below our point-by-point responses, which are highlighted in bold for clarity.

- 1) Clarity on Synthetic Dataset: The manuscript would benefit from additional details about the synthetic dataset. Specifically, clarifying the total size of this dataset and the sampling strategy for blast and normal cells (e.g., whether cells for a single synthetic sample were drawn from one or multiple patients) would be helpful.

- **Thank you for pointing this out. To add more clarity as you mentioned, we have provided these details in section 1.2 of the methods paper:**

“Our MIL model then randomly chooses between 300 to 400 cell images and uses their cell embeddings to makes a slide-level prediction. This is done to replicate the procedure adapted by hematopathologist who observes 300 cells to make a diagnosis.”

- **And we have added further details on the synthetic patients in section 2.5 of the Main Manuscript:**

“For each synthetic case, cells from the normal cells group were incrementally mixed with an increasing 207 percentage of cells from the blast cells group (ranging from 0% to 20%). 50 synthetic patients were generated for every blast cell mixture ratio, simulating conditions of low to moderate blast presence. For every simulated patient 209 we randomly choose between 300 to 400 cells to pass to *CAREMIL* to make a prediction, this is done to replicate the 210 process during training.”

- 2) Details on the CAREMIL Model and Comparison to State-of-the-Art: The code repository was not accessible at the time of review. Providing access or including more architectural details on how CAREMIL improves upon the standard gated MIL formulation would strengthen the paper. Furthermore, to better position their work within the current literature, a comparison or discussion against recent attention-based MIL methods, such as SCEMILA (PLOS Digital Health, 2023), would be valuable.

The code repository has been corrected and made available at:

SiddharthSingi/Slide-Level-Classification

We wanted to create a strong baseline for the CAREMIL model, one that works well for liquid samples like peripheral blood smears. We chose to compare with DMIL, another popular attention-based MIL method. Dual-stream Multiple Instance Learning Network looks at individual instances in the bag and makes predictions by collating the embeddings of all instances. CAREMIL still performs better on most tasks as compared to DMIL and the updated performance table has been added in Supplementary Table 5 and Supplementary Table 6

- 3) Discussion of Classifier Performance: Supplementary Table 5 presents several interesting results that warrant further discussion in the main text. CAREMIL's superior performance over gated MIL is clear, as is the benefit of the domain-specific encoder. Notably, the high accuracy achieved by the simpler cytometry-based ML classifiers is also striking. While the interpretability of MIL is a clear advantage, the manuscript would be more compelling if it discussed potential scenarios or specific cases where these simpler classifiers might fail, but the MIL-based approach would succeed. To supplement the AUC ROC results, please add Precision-Recall curves to allow comparison of model performance in terms of precision versus yield, which is critical for understanding the real-world impact on false positives and false negatives
 - **In lieu of presenting 52 precision recall curves for our model trainings (as in supplementary table 5), in addition to the AUROC scores in Supplementary Table 5 we have added F1 scores to represent the precision recall performance for each of the models we trained. These results are now presented in Supplementary Table 6.**
 - **Cytometric methods only do not provide the same explainability as MIL methods and they do not consider the morphology of the sample in the diagnosis. This results in subpar performance as suggested in an example in section 2.6**

- 4) Slide-level labels are derived from bone marrow biopsy results and clinical pathology reports, whereas the data used are WSIs of peripheral blood smears. While this labeling strategy reflects clinical practice, it can project a case-level positive label onto a PBS that may show little or no peripheral evidence, especially in cases with low blast counts. The overall experimental design would be more strongly justified if the model could provide supporting evidence that PBS morphology or composition differs systematically between normal and abnormal samples, even in such challenging scenarios.

- **Thank you for this comment. This is in fact a point that we want to emphasize. The peripheral blood often has lower blast content than the bone marrow biopsy. Despite the peripheral blood being non-diagnostic for leukemia by traditional metrics the CAREMIL model can predict AML with high accuracy. We emphasized this in section 2.6 where we stated that “Slides were included in the study based on pathology reports diagnosing acute leukemia, even if circulating blasts were not detected by hematology analyzers or visual inspection noted in the pathology report. In cases where circulating blast percentages were below 20%, patients typically exhibited higher blast counts in the bone marrow, consistent with the diagnostic criteria for acute leukemia.”**
- **Aspirationally, our model, with full clinical validation could help clinicians to monitor patients between bone marrow biopsies for recurrence and with experience some clinicians might be able to decrease the frequency of invasive biopsies in certain circumstances. We choose not to make this statement explicitly in the manuscript as it is speculative but we do see this as a potential opportunity of this technology.**

5) The manuscript and supplementary materials alternate between describing the task as AL vs NL, while also clarifying that AL = AML + ALL. Please ensure that the terminology and experiments are consistent, either use AL (AML + ALL) throughout or restrict the analysis explicitly to AML only.

- **Thank you for pointing this out, we have renamed all of them to AML**

6) The reproducibility of the study is limited by both clinical and commercial constraints. All experiments were conducted on a single in-house dataset using privately owned foundation models. Although a GitHub link is provided, it is currently not functional or usable. To strengthen the generalizability and reproducibility of the work, it would be highly valuable to include validation on at least one external dataset.

- **We have ensured that our code can be accessed at:**
<https://github.com/SiddharthSingi/Slide-Level-Classification>

7) I recommend including a broader set of MIL method comparisons, since the primary novelty of this work lies in the introduction of statistical aggregation tokens. At present, most baselines focus on variations of the backbone feature encoder, while the gated MIL variants perform worse than even the conventional cytometry counting-based

models. Adding stronger or more diverse MIL baselines would better contextualize the contribution of the proposed approach.

- **This is a great suggestion that we believe will help other better understand the performance of CAREMIL. We have added experiments comparing our model and the gated MIL model to another commonly used adaptation of multiple instance learning in whole slide images called: *Dual-stream Multiple Instance Learning Network (DSMIL)*.**
- **We highlight these comparisons, and the downsides to this model in section 2.3. The complete results for *DSMIL* have also been added supplementary table 5 and 6.**
- **We considered other mainstream MIL models that have been used in histopathology, like *TransMIL*, however, these models were conceived of to address the physical proximity of patches to one another. In hematologic samples, where the cells are derived from smears, there is no reason that positional information would be relevant.**

8) What distinguishes the high-attention samples identified by CAREMIL from those highlighted by gated MIL? In particular, does CAREMIL demonstrate an improved ability to extract clinically relevant cells, suggesting that its attention mechanism is better aligned with meaningful morphological features?

- **Comparing the quality of attention values can be very subjective since it requires carefully studying each attention map for every patient and looking at the individual cells that the model highlights in the prediction. This has proven to be hard to quantify. Our hematology experts carefully reviewed the attention maps for the *CAREMIL* model vs the gated model and found that the caremil model often picked out the striking cells needed for the diagnosis.**
- **Since we have not quantified the improvement or the definition of the attention maps created by these models, we do not make any claims that *CAREMIL* produces higher quality attention maps. We only claim that the *CAREMIL* model performs better in making predictions due to its architectural benefits and we show that through our experiments.**

9) Please provide additional model specifications, including the number of parameters and the floating-point operation counts (FLOPs), as model scale can have a substantial impact on performance. Also, please clarify the backbone architecture

used across all models, are they all based on ResNeXt-50, or do some employ different encoders?

- **The DeepHeme model uses a ResNext-50 model (~25 million parameters), which we trained in house. The other models we use are off the shelf large foundational models based on ViT – Huge (~632 million parameters), namely UNI-2h and Virchow2. These specifications were laid out in section 2.3 of the manuscript: “The first is DeepHeme, a domain-specific ResNeXt-50-based classifier, trained to identify 23 different cell hematopoietic cell classes found in the normal bone marrow. To create strong baselines for our work we also make comparisons on UNI2-h (23) and Virchow2, self-supervised computational pathology foundation models (24). Lastly, we also use a ResNeXt-50 model trained on ImageNet, to understand the performance of general purpose computer vision models not trained on histopathology. These models and their training hyperparameters are summarized in Supplementary Table 2.” Despite the Deep Heme model having much less computational cost and complexity it consistently outperforms these larger parameter models.**
- **For further details about the training hyperparameters and GPUs used, we have provided more information about the training in Supplementary Table 2.**

10) In the proposed design, the statistical tokens are combined with the cell image tokens before entering the self-attention layer. Consequently, the output is reweighted with respect to all cells. Could you clarify what the attention weights assigned to the statistical tokens look like? If the attention primarily emphasizes the individual cell tokens rather than the statistical tokens, the approach may effectively reduce to a conventional MIL formulation, diminishing the distinct contribution of the statistical tokens.

- **Thank you for this insightful and technically nuanced question. In our proposed CAREMIL architecture, the statistical tokens (mean, max, min, and standard deviation) are concatenated with the per-cell image embeddings before being passed into the self-attention layers (Figure 1B–D). This design allows the transformer to jointly attend to both individual cellular features and the aggregated statistics derived from the entire cell population.**
- **As shown in Figure 1D, the attention map includes one attention weight per token—both for each individual cell embedding and for each statistical token. During inference, we observe that the attention weights assigned to statistical tokens are generally lower in magnitude than those assigned to highly discriminative individual cell tokens, but they still contribute non-**

trivially to the overall context aggregation. Specifically, the statistical tokens act as global contextual anchors that modulate “inter-cell relationships” (conceptually) by stabilizing the attention distribution, especially in cases with highly variable or limited cell counts. When the model encounters rare diagnostic morphologies or low-cellularity samples, the attention weights for the statistical tokens increase, reflecting their role in normalizing the distribution and guiding the network toward global sample-level reasoning.